# Nickel-catalyzed reductive coupling of homoenolates and their higher homologues with unactivated alkyl bromides

Tingzhi Lin[1], Yuanyun Gu[1], Pengcheng Qian[1], Haixing Guan[1], Patrick J. Walsh [2✉] & Jianyou Mao [1✉]

The catalytic generation of homoenolates and their higher homologues has been a long-standing challenge. Like the generation of transition metal enolates, which have been used to great affect in synthesis and medicinal chemistries, homoenolates and their higher homologues have much potential, albeit largely unrealized. Herein, a nickel-catalyzed generation of homoenolates, and their higher homologues, via decarbonylation of readily available cyclic anhydrides has been developed. The utility of nickel-bound homoenolates and their higher homologues is demonstrated by cross-coupling with unactivated alkyl bromides, generating a diverse array of aliphatic acids. A broad range of functional groups is tolerated. Preliminary mechanistic studies demonstrate that: (1) oxidative addition of anhydrides by the catalyst is faster than oxidative addition of alkyl bromides; (2) nickel bound metallocycles are involved in this transformation and (3) the catalyst undergoes a single electron transfer (SET) process with the alkyl bromide.

[1] Technical Institute of Fluorochemistry (TIF), Institute of Advanced Synthesis, School of Chemistry and Molecular Engineering, Nanjing Tech University, 30 South Puzhu Road, Nanjing 211816, China. [2] Roy and Diana Vagelos Laboratories, Penn/Merck Laboratory for High-Throughput Experimentation, Department of Chemistry, University of Pennsylvania, 231 South 34th Street, Philadelphia, Pennsylvania 19104, USA. ✉email: pwalsh@sas.upenn.edu; ias_jymao@njtech.edu.cn

Homoenolates and their higher homologs represent important synthetic intermediates that have found broad applications in natural product synthesis and pharmaceutical sciences[1–3]. In comparison to ketones, which exhibit nucleophilic character upon deprotonation of the relatively acidic α-C–H's, the β- and γ-C–Hs are usually unreactive or, in the case of α,β-unsaturated derivatives, the β-carbon is electrophilic. Thus, with respect to α,β-unsaturated carbonyl compounds, the β-position of homoenolates exhibits umpolung reactivity[4–6]. Despite great progress in the generation and catalytic reactions of homoenolates[7,8], the demand for more straightforward approaches to these reactive species and their higher homologs remains high.

Early routes for stoichiometric generation of homoenolates include (a) β-deprotonation of camphenilone with KOtBu by Nickon and Lambert[9]; (b) TiCl$_4$-mediated ring opening of silyloxycyclopropanes by Nakamura and Kuwajima[10]; (c) Brønsted base-mediated metalation of allyl carbamates[11]; and (d) directed β-metalation of N,N-diisopropyl amides by Beak and colleagues[12,13], among others[14–20]. The generation and reactions of γ-metallocarbonyls, however, remains relatively unexplored[21].

Kuwajima, Nakamura, and their colleagues[22] pioneered catalytic reactions of palladium-bound homoenolates derived from the ring opening of cyclopropanols. The Pd-homoenolates coupled with aryl halides to furnish β-arylcarbonyl derivatives (Fig. 1a)[22]. However, this ring opening strategy cannot afford γ-metallocarbonyl analogs. In addition to arylations[23,24], other transformations such as acylation[25], benzylation[26], and allenylation[27] of palladium-homoenolates were also reported via ring opening of cyclopropanols. In the case of β-alkylation of homoenolates, only activated alkyl halides have been successfully employed[28–31]. For example, this year the Fu group developed a remarkable Ni-catalyzed coupling of racemic β-zincated amides with racemic propargylic bromides in a doubly stereoconvergent process to provide amides with high ee and dr (Fig. 1b)[32]. In addition to ring opening of cyclopropanols, directing group-assisted palladium-catalyzed β-C–H activations are an alternative strategy to generate and functionalize palladium-homoenolates of carboxylic acid derivatives[33–36]. More recently, a palladium-catalyzed functionalization of γ-C–H's of carboxylic acid derivatives was realized. This method employed 3,3-disubstituted butyric acid derivatives to circumvent the β-C–H functionalization pathway (Fig. 1c)[21,37]. To the best of our knowledge, the use of inexpensive nickel catalysts for the generation and functionalization of β-, or γ-bound carboxylic acid derivatives remains underdeveloped, especially related to coupling of these intermediates with alkyl groups[38,39]. To circumvent the use of unactivated alkyl halides, Rousseaux and colleagues[40] developed an elegant alkylation of homoenolates derived from cyclopropanols that involves use of redox active esters (Fig. 1d). In this work, redox active esters undergo in situ decarboxylation to provide alkyl radicals, leading to alkylated homoenolates[40]. To the best of our knowledge, catalytic alkylations of any type of homoenolate, or their higher homologs, with unactivated alkyl halides remains a challenge. This is likely due to the relatively slow oxidative addition of alkyl electrophiles and facile β-hydride elimination of the resulting intermediates[41–44].

We have been interested in developing new transformations of homoenolates[45,46] and cross-electrophile coupling of cyclic anhydrides with aryl triflates to afford aryl ketoacids[47]. We were inspired by Yamamoto's stoichiometric reactions of nickel(0) complexes with anhydrides and Rovis's stoichiometric nickel promoted transformation with cyclic anhydrides and ZnPh$_2$ (Fig. 2a)[48,49]. These latter reactions undergo insertion of nickel into the anhydride C–O bond followed by decarbonylation to generate a nickel homoenolate. Transmetallation from ZnPh$_2$ is followed by reductive elimination to form the C–C bond. Herein, we report a nickel-catalyzed alkylation of homoenolates and their higher homologs via decarbonylation of monocyclic anhydrides (Fig. 2b). The in situ-generated nickel homoenolate species are successfully coupled with unactivated alkyl bromides, producing a diverse array of valuable functionalized acids. In addition to alkylation of nickel bound homoenolates, γ-alkylation of butyric acid derivatives can also be performed under the same conditions. Compared with Martín's elegant carboxylation of alkyl bromides with CO$_2$, our strategy provides an alternative synthesis of functionalized acids[50,51]. It is noteworthy that functionalized aliphatic acids are popular structures in soaps, dyes, plastics, and many chemicals[52]. They are also useful precursors in coupling reactions[53–55].

**Fig. 1 Metal catalyzed functionalization of homoenolates and their higher homologs. a** Palladium-catalyzed arylations of cyclopropanol derivatives. **b** Ni-catalyzed doubly stereoconvergent coupling. **c** Palladium-catalyzed functionalization of γ-C–H's of carboxylic acid derivatives. **d** Ni-catalyzed alkylations with N-hydroxyphthalimides (NHPI).

## Results

**Reaction development and optimization.** To initiate the development of the nickel catalyzed alkylation of homoenolates, we chose succinic anhydride **1a** and 1-bromooctane **2a** as model substrates. Ni(COD)$_2$ and bipy were used as catalyst precursors in dimethylacetamide (DMA) at 80 °C for 12 h. As this is a cross-electrophile coupling reaction, a stoichiometric reducing agent is needed and zinc powder was chosen. After 12 h under these conditions, the desired decarbonylative product **3aa** was obtained in 63% assay yield (AY) (Table 1, entry 1, AY, determined by GC analysis of the unpurified reaction mixture with dodecane as internal standard). It should be noted that the non-decarbonylated γ-keto acid **3aa'** was not observed under these conditions. A survey of solvents found DMA was a better choice than *N,N*-dimethylformamide (DMF), tetrahydrofuran (THF), THF, or acetonitrile (AY 23–42%, entries 2–4, see Supplementary Information for details). To increase the AY of **3aa**, we next explored different nickel sources. Unfortunately, these nickel species afforded lower AY of **3aa** (18–52%, entries 5–8).

To improve the decarbonylative cross-coupling, we turned to examination of nitrogen-based ligands. Under the conditions of entry 1, the parent bipy outperformed all others examined (the complete optimization data is presented in the Supplementary Information). We next examined the effect of concentration on the AY. Increasing the concentration from 0.4 to 2.0 M had a beneficial impact, affording the desired product in 87% AY at 1.3 M (entries 1 and 9–12). When 1-chloro- or 1-iodohexane were used in place of 1-bromohexane, the AY were lower than 40% (see Supplementary Information). Therefore, our optimized reaction conditions are 1.5 equiv of anhydride **1a**, 1 equiv of alkyl bromide **2a**, 2 equiv of zinc powder, 10 mol % Ni(COD)$_2$, and 15 mol % bipy in DMA at 80 °C for 12 h.

**Substrate scope.** With the optimized reaction conditions in hand (Table 1, entry 11), we next explored the scope of the alkyl bromides. In general, a wide range of primary and secondary alkyl bromides participated in this transformation to afford the acid products in good to excellent yields with very good functional group tolerance (Table 2). We first chose the parent succinic anhydride **1a** the homoenolate precursor to couple with a variety of primary alkyl bromides. Simple long-chain 1-bromooctane **2a** underwent this coupling to afford **3aa** in 84% isolated yield. More challenging functionalized alkyl bromides bearing benzodihydrofuran, ester, imide, and ether functional groups were compatible under our conditions, generating the corresponding products **3ab**–**3af** in 68–91% yield.

We were also interested in using glutaric anhydride **1b**. Fortunately, glutaric anhydride underwent the decarbonylative alkylation smoothly, providing access to γ-alkylated acids under mild conditions. Glutaric anhydride reacted with a diverse array of primary alkyl bromides, including functionalized substrates, exhibiting good reactivity and furnishing the desired acids in

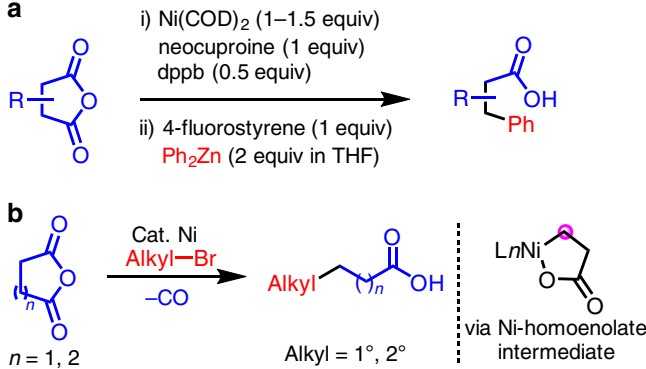

**Fig. 2 Nickel-mediated functionalization of cyclic anhydrides. a** Rovis's stoichiometric nickel complex with Ph$_2$Zn. **b** This work: Ni-catalyzed functionalization with alkyl bromides.

**Table 1 Optimization of the reaction conditions$^a$.**

| Entry | Solvent | Catalyst | Conc./M | AY$^b$ |
|---|---|---|---|---|
| 1 | DMA | Ni(COD)$_2$ | 0.4 | 63 |
| 2 | DMF | Ni(COD)$_2$ | 0.4 | 42 |
| 3 | THF | Ni(COD)$_2$ | 0.4 | 34 |
| 4 | CH$_3$CN | Ni(COD)$_2$ | 0.4 | 23 |
| 5 | DMA | NiI$_2$ | 0.4 | 52 |
| 6 | DMA | DME•NiBr$_2$ | 0.4 | 23 |
| 7 | DMA | NiCl$_2$•6H$_2$O | 0.4 | 27 |
| 8 | DMA | Ni(PPh$_3$)$_4$ | 0.4 | 18 |
| 9 | DMA | Ni(COD)$_2$ | 0.5 | 65 |
| 10 | DMA | Ni(COD)$_2$ | 1.0 | 70 |
| 11 | DMA | Ni(COD)$_2$ | 1.3 | 87 |
| 12 | DMA | Ni(COD)$_2$ | 2.0 | 86 |

$^a$Reactions conducted on a 0.1 mmol scale under argon atmosphere.
$^b$AY, assay yield, determined by GC analysis.

**Table 2 Substrate scope with monocyclic anhydrides and alkyl bromides[a].**

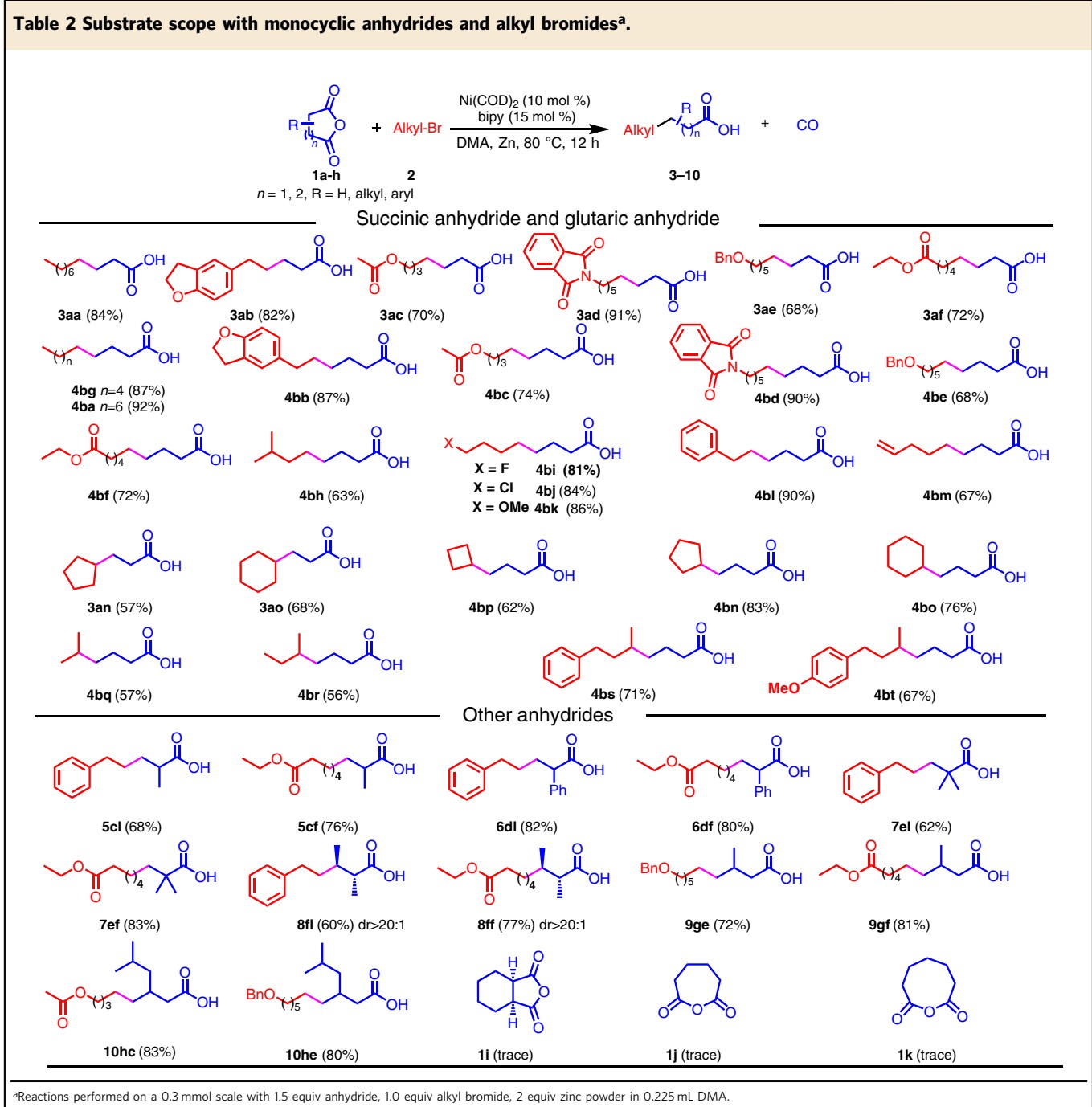

68–92% yields (**4ba–4bg**). β-Branched alkyl bromides are expected to generate nickel alkyl intermediates that tend to undergo relatively fast β-hydride elimination[56,57]. Nonetheless, isobutyl bromide was compatible under the standard conditions, furnishing **4bh** in 63% yield. Furthermore, alkyl bromides bearing halogens, such as fluoro and chloro reacted with excellent chemoselectivity, affording the corresponding products in 81 and 84% yield, respectively. Alkyl bromides containing ether, homobenzyl, and double bonds reacted efficiently, affording the products in 67–90% yields.

In addition to primary alkyl bromides, a variety of secondary alkyl bromides were compatible with the standard conditions. In general, both cyclic and acyclic secondary alkyl bromides, such as cyclopentyl-, cyclohexyl-, cyclobutyl-, isopropyl-, and *sec*-butyl- bromide, exhibited good reactivity, affording

β- and γ-functionalized propionic and butyric acids, respectively (**3an**, **3ao**, and **4bn–4bt**, 56–83%). Unfortunately, when tertiary bromides were subjected to coupling with anhydrides, no products were obtained under the standard conditions.

We next explored the scope of the unsymmetrical anhydrides. It is observed that unsymmetrical succinic anhydrides such as 2-methylsuccinic anhydride **1c**, 2-phenylsuccinic anhydride **1d** and 2,2-dimethylsuccinic anhydride **1e** could couple with **2l** and **2f** affording β-functionalized acids in 68–83% yield. It is noteworthy that the oxidative addition and decarbonylation processes took place at the less sterically hindered side of the anhydride. Disubstituted anhydrides, such as *cis*-2,3-dimethylsuccinic anhydride **1f** is a competent coupling partner, affording **8fl** and **8ff** in 60% and 77% yield, respectively. It should be noted that **1f** led to the corresponding products with high stereochemical fidelity

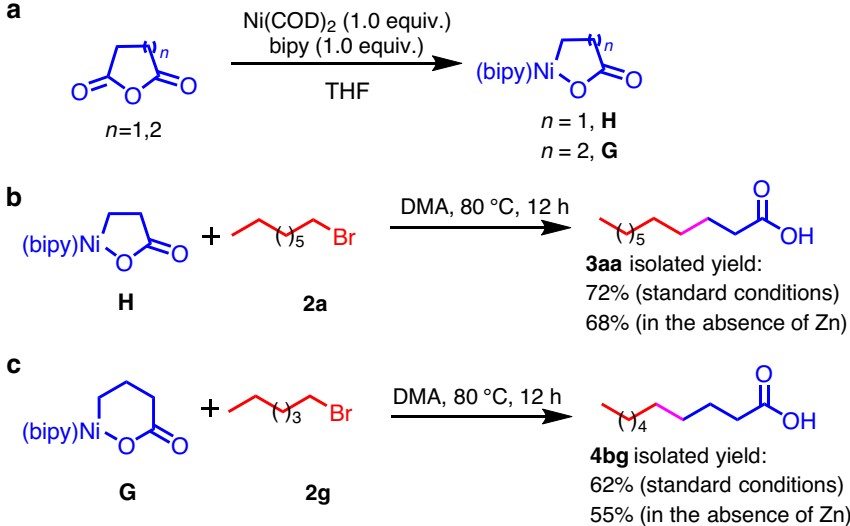

**Fig. 3 Scale up the transformation to 10.0 mmol.** 1-Bromo-4-chlorobutane reacted with **1b** to give the desired product **4bj** with 87% yield.

**Fig. 4 Preliminary mechanistic studies. a** Preparation of Ni-homoenolate and its higher homolog. **b** Alkylation of the independently synthesized homoenolate **H**. **c** Alkylation of the independently synthesized **G** with alkyl bromides.

(>20 : 1 dr). Furthermore, methyl and isobutyl substituted glutaric anhydrides **1g** and **1h** afforded γ-cross-coupled products in 72–83% yield. Unfortunately, bicyclic anhydrides, such as *cis*-1,2-cyclohexanedicarboxylic anhydride **1i** and larger ring-sized anhydrides (**1j** and **1k**), gave only trace coupling products under the standard conditions.

To test the scalability of this transformation, 15.0 mmol of glutaric anhydride **1b** was coupled with 10.0 mmol of 1-bromo-4-chlorobutane **2j** under the standard conditions. The decarbonylative cross-coupled product **4bj** was isolated in 87% yield (1.55 g) (Fig. 3).

**Mechanistic studies.** Preliminary mechanistic studies demonstrated that nickel homoenolates are intermediates in this catalytic system. Ni-homoenolate **H** and its higher homolog **G** were synthesized independently[58,59] (Fig. 4a). Exposure of **H** or **G** to alkyl bromides in the presence or absence of reducing agent generated **3aa** and **4bg** in good yields (Fig. 4b, c). These results demonstrate that Ni-bound homoenolate **H** and its derivative **G** are viable intermediates in these transformations. We have ruled out the possibility of cyclic anhydrides undergoing CO deinsertion followed by β-hydride elimination to generate acrylate intermediates and conjugate addition of radicals to these intermediates (see Supplementary Information). Notably, the stoichiometric reactions were successful in the absence of zinc powder (the source of electrons for this process may be the Ni(I) product reducing the alkyl bromide; see the catalytic cycle discussed below)[60,61]. These results lead to the hypothesis that zinc powder acts as a reductant to nickel to make these reactions catalytic[62–67].

To explore the selectivity in the oxidative addition to (L)Ni⁰, we examined the relative reactivities of glutaric anhydride (**1b**) and 1-bromohexane (**2g**) using stoichiometric Ni(COD)₂ and bipy (Table 3) in a study that was inspired by the Weix group's report of oxidative addition with nickel using aryl and alkyl halides[68]. After these reagents were stirred at room temperature for several hours, the reaction was quenched with 1 M HCl. The relative reactivities were determined by the loss of each starting material and formation of **G** and **4bg**[68] (Table 3). We found anhydride **1b** was consumed much faster than alkyl bromide **2 g**. These data indicate that the anhydride undergoes oxidative addition faster than the alkyl bromide in the presence of Ni (COD)₂ and bipy, and that **G** is a likely intermediate in the catalytic reaction.

To further probe the reaction mechanism, radical trap and radical clock experiments were employed (Fig. 5a, b)[66,69]. Under otherwise standard conditions with alkyl bromide **2l**, when the radical scavenger TEMPO was added, product formation was suppressed. If the alkyl bromide is activated through a radical mechanism, ring-opened products would be expected with cyclopropylmethyl bromide (**2u**). In the event, **2u** led primarily to unsaturated acid **4bu'** (52% AY), arising from rapid ring opening of the cyclopropyl-methyl radical to the homoallylic radical.

Inspired by Weix's 5-exo-trig experiment to evaluate ring-closed vs. ring-opened ratios as a function of catalyst loading[68], we chose glutaric anhydride (**1b**) and 5-hexenyl bromide (**2v**) to examine the ratio of decarbonylative product **4bv** and cyclized product **4bv'**. By changing the catalyst loading of Ni(COD)₂ from 5 mol% to 40 mol%, the ratio of **4bv/4bv'** was observed to increase linearly (Fig. 5c).

Finally, we probed the possibility of an organozinc intermediate in this cross-coupling, potentially formed from reaction of Zn⁰ with the alkyl bromide. Thus, we independently synthesized organozinc reagent **J** and subjected it to the metallocycle **G** under catalytically relevant conditions (Fig. 5d). No conversion to cross-coupled product was observed. Taken

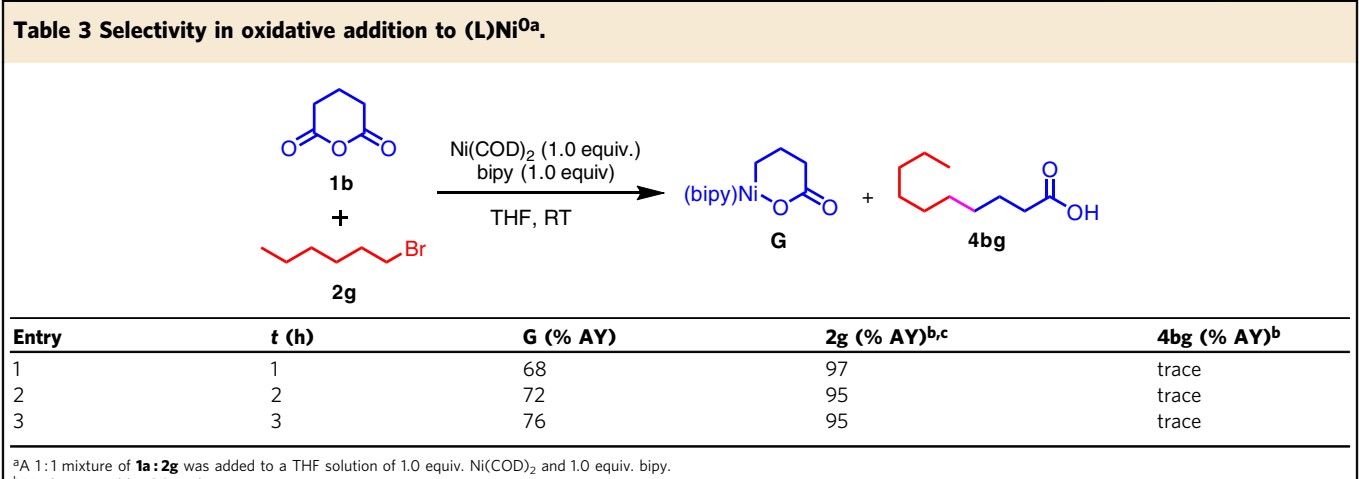

**Table 3 Selectivity in oxidative addition to (L)Ni[0a].**

| Entry | t (h) | G (% AY) | 2g (% AY)[b,c] | 4bg (% AY)[b] |
|-------|-------|----------|----------------|----------------|
| 1 | 1 | 68 | 97 | trace |
| 2 | 2 | 72 | 95 | trace |
| 3 | 3 | 76 | 95 | trace |

[a]A 1:1 mixture of **1a:2g** was added to a THF solution of 1.0 equiv. Ni(COD)$_2$ and 1.0 equiv. bipy.
[b]AY determined by GC analysis.
[c]Remaining **2g**, determined by GC analysis.

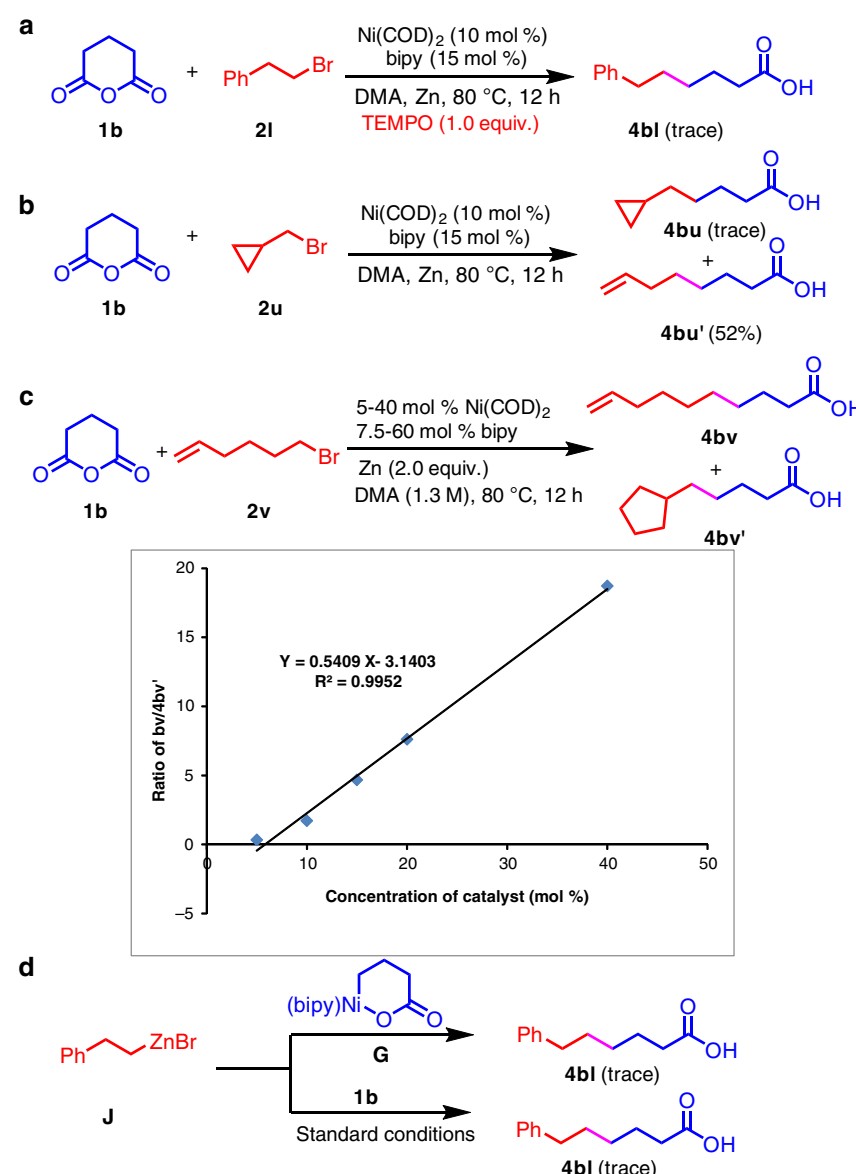

**Fig. 5 Mechanistic studies. a** Inhibition of the reaction in the presence of TEMPO. **b** Radical clock reaction leads to ring opening, suggestive of radical intermediates. **c** Dependence of the ratio of **4bv/4bv'** on catalyst concentration, supporting a radical mechanism. **d** Demonstration that organozinc **J** is not a viable reaction intermediate.

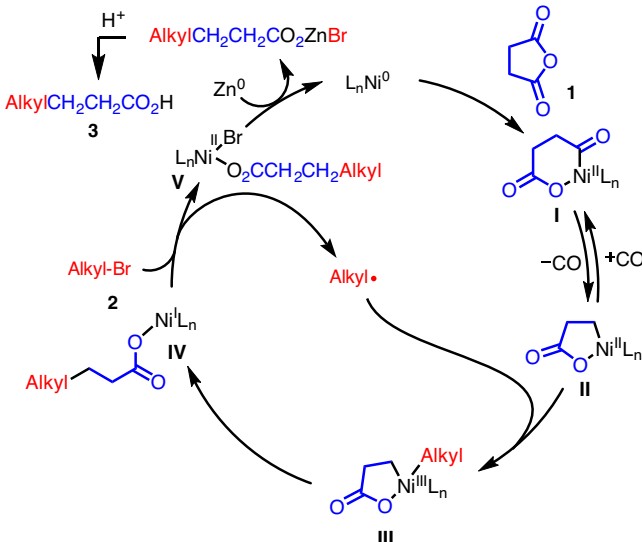

**Fig. 6 Mechanistic hypothesis.** Oxidative addition of anhydride (**1**) is followed by loss of carbon monoxide to generate homoenolate **II**. Oxidative capture of the alkyl radical formed Ni(III), which reductively eliminates to form Ni(I) intermediate **IV**. The Ni(I) intermediate is proposed to reduce the alkyl bromide **2** to furnish the alkyl radical and Ni(II) intermediate **V**. Zinc reduction of Ni(II) reforms Ni(0) and produces a zinc carboxylate, closing the catalytic cycle. Upon workup, the zinc carboxylate is protonated to give the observed carboxylic acid product **3**.

together, these experiments support single electron transfer (SET) processes[70] and discount an organozinc intermediate.

Grounded in the experimental results presented above, a plausible mechanism is proposed (Fig. 6). Ni[0] oxidatively adds cyclic anhydride **1** giving a Ni[II] species (**I**)[71–74]. Next, decarbonylation of **I** with loss of CO forms the primary $C(sp^3)$-Ni bond[75] in **II**. The next step is generation of the alkyl radical via SET. This could take place from Ni(0) or Ni(I). We postulate that an intermediate Ni(I) performs the SET, because Table 3 suggests that reaction of Ni(0) with alkyl bromide is slow relative to oxidative addition of the anhydride. The alkyl radical is oxidatively trapped to provide the Ni[III] complex (**III**). Reductive elimination of intermediate **III** generates the C–C bond as well as the reactive Ni[I] carboxylate species **IV**. As noted above, this Ni(I) species is proposed to reduce the alkyl bromide via SET giving BrNi(II)(carboxylate) (**V**) and an alkyl radical. Finally, the Ni[II] product **V** is reduced by Zn[0] to regenerate Ni[0] and the zinc carboxylate. Acidic workup liberates the acid coupling product, **3**.

## Discussion

We have developed a nickel-catalyzed generation of homo-enolates and their higher homologs via decarbonylation of monocyclic anhydrides. The in situ-formed nickel homoenolate derivatives were successfully coupled with unactivated alkyl bromides, generating a diverse array of functionalized aliphatic acids. Preliminary mechanistic studies demonstrated that the nickel(0) catalyst selectively reacts with anhydrides over alkyl bromides leading to the key homoenolates. Radical clock and related studies indicate that organozinc intermediates are unlikely and point to the involvement of alkyl radical species. Key advantages of this method include excellent functional group compatibility, mild reaction conditions, and avoidance of pre-functionalized organometallic reagents. This method enables construction of valuable carboxylic acids. Further studies using anhydrides as homoenolate precursors are ongoing in our team.

## Methods

**General procedure for the synthesis of aliphatic acids.** To an oven-dried microwave vial (10 mL) equipped with a stir bar (10 × 5 mm) was added Ni(COD)$_2$ (8.3 mg, 0.03 mmol) and bipy (7.0 mg, 0.045 mmol) under an argon atmosphere inside a glove box at 25 °C. Next, 0.225 mL of dry DMA was added via syringe to give a purple solution. After the catalyst/ligand solution was stirred for 1 h at 25 °C inside the glove box, Zn powder (39.2 mg, 0.6 mmol, 2.0 equiv) was added to the reaction vial followed by the monocyclic anhydride (0.45 mmol, 1.5 equiv) and alkyl bromide (0.3 mmol, 1.0 equiv). The microwave vial was sealed with a cap containing a rubber septum and removed from the glove box. The reaction mixture was stirred (~1000 r.p.m.) at 80 °C for 12 h. The resulting gray solution was cooled to room temperature, quenched by the addition of five drops of water via syringe through the septum and then the vial opened to air. The reaction mixture was passed through a short flash column chromatography in silica gel (200–300 mesh) and rinsed with 5 mL of ethyl acetate to afford a yellow solution. The solvent and volatile materials were removed by rotary evaporator. The crude residue was purified by flash column chromatography in silica gel to yield the corresponding product.

## Data availability

Detailed experimental procedures and characterization of compounds can be found in the Supplementary Information. The authors declare that all other data supporting the findings of this study are available within the article and Supplementary Information files, and also are available from the corresponding authors.

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

## Acknowledgements

We acknowledge the National Natural Science Foundation of China (21801128 and 22071107 to J.M.), Natural Science Foundation of Jiangsu Province, China (BK20170965 to J.M.), and Nanjing Tech University (39837112) for financial support. P.J.W. thanks the US National Science Foundation (CHE-1902509).

## Author contributions

T.L. performed most of the experiments and mechanistic study with the help of Y.G., P.Q., and H.G. The project conceived J.M. and T.L. with help from P.J.W. The project was directed by J.M. and the manuscript was written by T.L., J.M., and P.J.W.

## Competing interests

The authors declare no competing interests.
