## [Peer Review File · Nature Communications]

REVIEWER COMMENTS

Reviewer #1 (Remarks to the Author):

This report by Walsh, Mao and coworkers describes the Ni-catalyzed reductive coupling of alkyl bromides with 5- and 6-membered cyclic anhydrides to form β - and γ -alkylacid products. In the case of 5-membered anhydrides, the reaction proceeds via a Ni homoenolate intermediate, which is intriguing since Ni homoenolates have not been widely investigated as synthetic intermediates. Overall, the disclosed procedure is useful for accessing the products described and the mechanistic studies which support the authors' proposed radical chain mechanism are of significant value. The experiment presented in Table 3 is valuable as it shows the propensity of Ni(0) to undergo oxidative addition with the C(sp²)-X electrophile rather than the C(sp³)-X electrophile. This experiment is complementary to a similar one performed by Weix and coworkers in the context of alkyl halide and aryl halide cross-electrophile coupling. The authors provide the proper reference at that point in the manuscript (ref. 63), though it would be useful to have a note mentioning that this type of experiment was originally performed by Weix and coworkers. I would also recommend adding references to recent reviews by Diao and coworkers which discuss the relative ease of oxidative addition of Ni(0) and Ni(I) with C(sp²)-X species and C(sp³)-X species (Trends Chem. 2019, 1, 830–844; Acc. Chem. Res. 2020, 53, 906–919).

The radical chain mechanism proposed by the authors is reasonable and supported by experiments in Fig 5b and 5c. The 5-exo-trig experiment developed by Weix and coworkers (JACS 2013, 135, 16192–16197) to evaluate ring-closed versus ring-opened ratios as a function of catalyst loading would further support this mechanism, though this experiment may be left for a later study of these systems (especially given the current situation and any lab shutdown that may be in effect due to COVID-19).

Overall, as stated above, this is a nice contribution to the field of Ni homoenolate chemistry and reductive cross-electrophile couplings and should be considered for publication in Nature Communications after minor revisions.

Additional comments that should be addressed prior to publication:

1. The content of Fig. 1 is confusing, since it is not obvious how NHC chemistry is directly related to this manuscript. Yes, NHC catalysis can act with homoenolate-like reactivity, but these species are not considered the same type of reagent as metal homoenolates, which is the topic of this manuscript. This reviewer believes Fig. 1 will affect readers' impression of the manuscript in a way that does not complement its elegant chemistry, and it should be removed. Additionally, the paragraph beginning with "The catalytic generation of homoenolates is also promising, but remains underdeveloped" is not relevant to metal homoenolate chemistry. It should be changed to say that these NHC-based intermediates can act as homoenolate synthons and yield products similar to those generated using metal homoenolate chemistry. Or, the paragraph could be removed altogether. Instead of NHC chemistry, it would be more appropriate to showcase one of the first reports of catalytic metal homoenolate formation, which was the Pd-catalyzed arylation of TMS-protected cyclopropanone hemiacetals with aryl triflates as reported by Nakamura, Kuwajima and coworkers (ref 29).

2. The statement "In the case of β -alkylation of homoenolates, only activated allylic halides have been successfully employed" should be revised, since Dai and coworkers have functionalized Cu homoenolates with α -bromocarbonyl derivatives (Org. Lett. 2015, 17, 6074–6077; ACS Catal. 2018, 8, 5907–5914). The important report by Matsubara and coworkers on the Cu-catalyzed allylation of Zn homoenolates should also be referenced (Chem. Lett. 2007, 36, 164–165).

3. Important precedence for the chemistry in this manuscript is the authors' work on the Ni-catalyzed reductive coupling of aryl triflates with cyclic anhydrides (Org. Lett. 2018, 20, 1191–1194). It is not clear why this report is not referenced, since it is synthetically related to the chemistry presented in this manuscript and should be mentioned. This will not detract from the mechanistic implications and impact of this manuscript.

4. In the reaction of 5-membered cyclic anhydrides to form Ni homoenolate intermediates, do the authors have any evidence that β -hydride elimination does not occur after CO elimination? This

would generate the acrylate intermediate which could react in a radical conjugate addition with the alkyl bromide under reductive conditions (Tetrahedron Lett. 1989, 30, 689–692). The authors recognize that β -hydride elimination is a challenge in the cross-coupling of alkyl coupling partners early on in the manuscript, saying that for alkyl coupling there is “facile β -hydride elimination of the resulting intermediates.” The authors should at least discuss (or ideally provide experimental evidence if possible) why they think this mechanism is not at play to a significant degree, which would help support the role of the Ni homoenolate and its stability under these reaction conditions.

5. Have control experiments (no Ni, no Zn, no ligand) been performed? Have other reductants besides Zn been tried? I couldn't find these in the SI. These should be added in the revised manuscript.

6. It appears that decarbonylation is an easy process, even occurring at room temperature (based on Table 3). I was a little surprised by this since previous work by Rovis has shown that stoichiometric Ni is required for decarbonylation to occur as the Ni actually sequesters the CO in the process. Can the authors comment on the fate of CO in their reaction? Is it trapped by Ni? Should a word of caution be added to the experimental procedure (and manuscript text) that Ni-CO complexes may be formed in situ?

7. It's a shame that the reaction with bicyclic anhydrides did not work. Do the authors have a working hypothesis as to why this is? Obviously, the products of these reactions would be very interesting, especially if stereochemistry could be controlled in the process.

8. In Figure 6c (bottom arrow), is the starting material 1b here (instead of 1a) or is there a mistake in the product structure?

Reviewer #2 (Remarks to the Author):

The manuscript by Walsh, Mao et al. describes a nickel-catalyzed decarbonylative cross-electrophile coupling between cyclic anhydrides and alkyl bromides to afford alkylcarboxylic acids. The reaction is promoted using a catalytic system comprised of Ni(cod)₂, bipyridine and zinc dust in DMA at 80 °C, affording the desired acid products in moderate to good yields. The reaction is applicable to succinic and glutaric anhydrides as well as to primary and secondary alkyl bromides with tolerance to several functional groups. Mechanistic experiments indicated the intermediacy of a nickelalactone (“nickel homoenolate” in the case of a five-membered cycle) and an alkyl radical, while eliminating possible involvement of an alkylzinc bromide. Based on the experimental results and the known organometallic chemistry of nickel, a catalytic cycle featuring C-O oxidative addition of the anhydride to Ni(0) and subsequent decarbonylation as well as single electron transfer from Ni(I) to the alkyl bromide was proposed.

The nickel(0)-mediated activation of cyclic anhydride and subsequent decarbonylation have been known for some time, as demonstrated by the previous works by Yamamoto and Rovis. That said, the present work is notable in that it has successfully translated this stoichiometric reactivity into a productive catalytic process. The reaction would be useful as a method for the synthesis of carboxylic acids from readily available cyclic anhydrides and alkyl bromides. Publication of this manuscript in Nature Communications may be positively considered, while the authors are suggested to address the following points.

1. The authors focus on the chemistry of homoenolates in the Introduction, devoting a lot of sentences and figures to discuss seminal works on homoenolates, NHC-derived homoenolates (Fig 1), metal homoenolates derived from cyclopropanol derivatives (Fig 2a,c), and organozinc homoenolates (Fig 2b). However, the reviewer feels that this introduction is not fully relevant, and would urge the authors to reconsider how the present work should be introduced. This is because the organonickel species generated from glutaric anhydride, while being referred to as “higher homologue” of homoenolate, has little to do with the chemistry discussed in the Introduction. Focusing solely on homoenolate in the Introduction looks irrelevant, especially given the substantial weight of glutaric anhydride (and its derivative) in the reaction scope. In fact, more than 70% of the products in Table 2 are derived from glutaric anhydride derivatives.

The reviewer believes it would be possible to compose a more relevant introduction to this work from different contexts, for example, (Ni-catalyzed) C-O activation, decarbonylation, cross-electrophile coupling, carboxylic acid synthesis, and so on.

2. Page 4, "We have been interested in developing new transformations of homoenolates." The reviewer cannot see the relevance of one of the references cited for this statement (ref 42 by Wan et al). A clarification is necessary.

3. The authors are suggested to explore substituted succinic anhydrides to better clarify the scope and limitation, and (if the reaction worked) the regioselectivity of C-O cleavage and C-C coupling.

4. Does the reaction work using alkyl iodides or chlorides instead of alkyl bromides? Relevant comments would better clarify the scope of this reaction.

5. Page 7, "...10.0 mmol of glutaric anhydride 1b was coupled with 1-bromo-4-chlorobutane 2j..." This should be corrected, as 15 mmol of glutaric acid and 10 mmol of alkyl bromide were used according to the Supporting Information. In addition, Fig 4 also needs correction because "4b 10.0 mmol 1.55 g 87%" below the product structure is misleading.

Reviewer #3 (Remarks to the Author):

This manuscript describes a Ni-catalyzed decarbonylative alkylation reaction of cyclic anhydrides. Upon oxidative addition to a cyclic anhydride, Ni undergoes decarbonylation to yield a key cyclic Ni-homoenolate intermediate that is subsequently alkylated to give beta- and gamma- alkylated carboxylic acids. Analogous reactions have been reported by Yamamoto and Rovis (ref. 43, 44) using stoichiometric Ni and Zinc reagents. The authors used Zn as the terminal reductant to accomplish Ni turnover. The authors claim catalytic generation of homoenolate remains underdeveloped. In fact, the vast pool of reports on Csp³-H activation of carboxylic acids and their derivatives were ignored. The products involved in this manuscript can be readily prepared using beta- and gamma-Csp³-H alkylation protocols, on a much broader substrate scope. Therefore, novelty and synthetic potential of the current work is capped. The work might be publishable if the authors can go beyond six-membered cyclic anhydride to large ring size, which is surely outside of the scope of most C-H functionalization methods. Alternatively, if the authors can accomplish enantioselective alkylation using branched, racemic/meso anhydrides and chiral ligands, the manuscript would be significantly strengthened. Based on recent reports by Fu, it should be accomplishable.

Other issues

1. The introduction part needs major revision to include progress in the C-H activation paradigm and clearly show advantages of the current work.

2. Fig. 7, mechanistic cycle, is it possible that Zn reduces intermediate II to Ni(I) and the corresponding Ni(I) undergoes oxidative addition to alkyl-Br? This alternative pathway also gives III and radical mechanism can still operate.

3. Page 10, last paragraph, there are two "we" in the second sentence.

4. Substrates with ring size beyond glutaric anhydride need to be commented.

Below please find our reply to the reviewer's comments.

COMMENTS TO AUTHOR:

Reviewer 1: This report by Walsh, Mao and coworkers describes the Ni-catalyzed reductive coupling of alkyl bromides with 5- and 6-membered cyclic anhydrides to form β - and γ -alkyl acid products. In the case of 5-membered anhydrides, the reaction proceeds via a Ni homoenolate intermediate, which is intriguing since Ni homoenolates have not been widely investigated as synthetic intermediates. Overall, the disclosed procedure is useful for accessing the products described and the mechanistic studies which support the authors' proposed radical chain mechanism are of significant value. The experiment presented in Table 3 is valuable as it shows the propensity of Ni(0) to undergo oxidative addition with the C(sp²)-X electrophile rather than the C(sp³)-X electrophile. This experiment is complementary to a similar one performed by Weix and coworkers in the context of alkyl halide and aryl halide cross-electrophile coupling. The authors provide the proper reference at that point in the manuscript (ref. 63), though it would be useful to have a note mentioning that this type of experiment was originally performed by Weix and coworkers. I would also recommend adding references to recent reviews by Diao and coworkers which discuss the relative ease of oxidative addition of Ni(0) and Ni(I) with C(sp²)-X species and C(sp³)-X species (Trends Chem. 2019, 1, 830–844; Acc. Chem. Res. 2020, 53, 906–919).

Response: First, we appreciate the reviewer's suggestions. We modified the sentence "In order to explore the selectivity in the oxidative addition to (L)Ni⁰, we examined the relative reactivities of glutaric anhydride (**1b**) and 1-bromohexane (**2g**) using stoichiometric Ni(COD)₂ and bipy (Table 3) in a study that was inspired by the Weix group's report of oxidative addition with nickel using aryl and alkyl halides.⁶⁸." Also, prof. Diao and coworkers work have been added in the revised manuscript as refs 71 and 72 .

Reviewer 1: The radical chain mechanism proposed by the authors is reasonable and supported by experiments in Fig 5b and 5c. The 5-exo-trig experiment developed by Weix and coworkers (JACS 2013, 135, 16192–16197) to evaluate ring-closed versus ring-opened ratios as a function of catalyst loading would further support this mechanism, though this experiment may be left for a later study of these systems (especially given the current situation and any lab shutdown that may be in effect due to COVID-19).

Response: We performed the 5-exo-trig experiment (shown below), which is a reasonable method to support our mechanism. We chose glutaric anhydride **1b** and 6-bromohex-1-ene **2v** as model substrates, Ni(COD)₂ and bipy as catalyst precursors, Zn

powder as reductant. After 12 h at 80°C the desired decarbonylative product **4bv** and cyclized product **4bv'** were obtained in 62% yield as a mixture. Then we tried different catalyst concentrations from 5 mol % to 40 mol % and the corresponding results are shown below. Error bars are the standard deviation of the data used for the plot. Linear fit: $Y = 0.5409 X - 3.1403$; $R^2 = 0.9952$.

These results have been added in Fig 5c.

Additional comments:

Reviewer 1: The content of Fig. 1 is confusing, since it is not obvious how NHC chemistry is directly related to this manuscript. Yes, NHC catalysis can act with homoenolate-like reactivity, but these species are not considered the same type of reagent as metal homoenolates, which is the topic of this manuscript. This reviewer believes Fig. 1 will affect readers' impression of the manuscript in a way that does not complement its elegant chemistry, and it should be removed. Additionally, the paragraph beginning with "The catalytic generation of homoenolates is also promising, but remains underdeveloped" is not relevant to metal homoenolate chemistry. It should be changed to say that these NHC-based intermediates can act as homoenolate synthons and yield products similar to those generated using metal homoenolate chemistry. Or, the paragraph could be removed altogether. Instead of NHC chemistry, it would be more appropriate to showcase one of the first reports of catalytic metal homoenolate formation, which was the Pd-catalyzed arylation of TMS-protected cyclopropanone

hemiacetals with aryl triflates as reported by Nakamura, Kuwajima and coworkers (ref 29).

Response: We have removed the sentence “The first catalytic NHC induced generation of” and the introduction with respect to NHC homoenolate in the revised manuscript. We have focused on the catalytic generation of metal homoenolates and their higher homologues instead of NHC chemistry.

Reviewer 1: The statement “In the case of β -alkylation of homoenolates, only activated allylic halides have been successfully employed” should be revised, since Dai and coworkers have functionalized Cu homoenolates with α -bromocarbonyl derivatives (ACS Catal. 2018, 8, 5907–5914 and Org. Lett. 2015, 17, 6074–6077). The important report by Matsubara and coworkers on the Cu-catalyzed allylation of Zn homoenolates should also be referenced (Chem. Lett. 2007, 36, 164–165).

Response: We modified the text to read: “In the case of β -alkylation of homoenolates, only activated *alkyl halides* have been successfully employed” in the revised manuscript. Refs for functionalized Cu homoenolates with α -bromocarbonyl derivatives by Dai and coworkers were added in the revised manuscript (refs 29 and 30). Also, Matsubara and coworkers’ work on the Cu-catalyzed allylation of Zn homoenolates was added in the early routes for stoichiometric generation of homoenolates (ref 18).

Reviewer 1: Important precedence for the chemistry in this manuscript is the authors’ work on the Ni-catalyzed reductive coupling of aryl triflates with cyclic anhydrides (Org. Lett. 2018, 20, 1191–1194). It is not clear why this report is not referenced, since it is synthetically related to the chemistry presented in this manuscript and should be mentioned. This will not detract from the mechanistic implications and impact of this manuscript.

Response: We are not sure how this reference got dropped from the submitted version. Sorry for our oversight. We have been interested in developing new transformations of homoenolates and desymmetrizing cross-electrophile coupling of cyclic anhydrides. Our reference on the Ni-catalyzed reductive coupling of aryl triflates with cyclic anhydrides was added in the revised manuscript (ref 47).

Reviewer 1: In the reaction of 5-membered cyclic anhydrides to form Ni homoenolate intermediates, do the authors have any evidence that β -hydride elimination does not occur after CO elimination? This would generate the acrylate intermediate which could react in a radical conjugate addition with the alkyl bromide under reductive conditions (Tetrahedron Lett. 1989, 30, 689–692). The authors recognize that β -hydride

elimination is a challenge in the cross-coupling of alkyl coupling partners early on in the manuscript, saying that for alkyl coupling there is “facile β -hydride elimination of the resulting intermediates.” The authors should at least discuss (or ideally provide experimental evidence if possible) why they think this mechanism is not at play to a significant degree, which would help support the role of the Ni homoenolate and its stability under these reaction conditions.

Response: We used the 5-membered cyclic anhydride **1a** and 1-bromooctane **2a** to determine whether β -hydride elimination was occurring after CO elimination. Under the standard conditions, this reaction provided 86% yield of **3aa**. When GC and GCMS were used to monitor the reaction mixture, we did not find the acrylic acid **A** or acrylic lactone **B**. Next, we used acrylic acid **A** instead of succinic anhydride **1a** as starting material under the standard conditions. We did not detect product **3aa** in this reaction mixture. We hypothesize that β -hydride elimination is diminished in the presence of carbon monoxide, which can occupy a coordination site on the nickel and block β -H elimination. Taken together, these results suggest that β -hydride elimination after CO deinsertion did not occur.

Reviewer 1: Have control experiments (no Ni, no Zn, no ligand) been performed? Have other reductants besides Zn been tried? I couldn't find these in the SI. These should be added in the revised manuscript.

Response: An 86% yield of **3aa** under the standard conditions was obtained when succinic anhydride (**1a**) and 1-bromooctane (**2a**) were employed. However, when the reaction was carried out without nickel, no cross-coupled product **3aa** was observed. In the absence of ligand or reductant, only low yields of **3aa** was formed. Replacement of zinc powder reductant with manganese powder give a lower yield of **3aa**. We have added these experiments to the revised Support Information.

Entry	Ni source	Ligand	Reductant	Yield ^a (%)
1	Ni(COD) ₂	bipy	Zn	86
2	none	bipy	Zn	0
3	Ni(COD) ₂	none	Zn	trace
4	Ni(COD) ₂	bipy	none	<5
5	Ni(COD) ₂	bipy	Mn	68
6	Ni(COD) ₂	bipy	Mg	25

^aIsolated yields.

Reviewer 1: It appears that decarbonylation is an easy process, even occurring at room temperature (based on Table 3). I was a little surprised by this since previous work by Rovis has shown that stoichiometric Ni is required for decarbonylation to occur as the Ni actually sequesters the CO in the process. Can the authors comment on the fate of CO in their reaction? Is it trapped by Ni? Should a word of caution be added to the experimental procedure (and manuscript text) that Ni-CO complexes may be formed in situ?

Response: For the oxidative addition of cyclic anhydrides, we referenced Yamamoto's work (*Bull. Chem. Soc. Jpn.* **1984**, *57*, 2741-2747, see below). Compound **H** is considered to be formed through oxidative addition of succinic anhydride **1a** to Ni(0) giving six-membered product **A** followed by its decarbonylation. Carbon monoxide thus released is trapped by Ni(0) complexes to afford Ni(CO)₂(bipy) (confirmed by IR, *Bull. Chem. Soc. Jpn.* **1981**, *54*, 2161-2168; *J. Chem. Soc.* **1953**, 2670-2673). Under our standard reaction conditions, Ni(COD)(bipy) reacts with CO to form the off-cycle intermediate Ni(CO)₂(bipy). However, we anticipate at elevated temperature (>80 °C), Ni(CO)₂(bipy) can reenter the catalytic cycle, presumably via thermal dissociation of the carbonyl ligands (*J. Am. Chem. Soc.* **2019**, *141*, 17322–17330; *J. Am. Chem. Soc.* **2012**, *134*, 13573-13576).

Reviewer 1: It's a shame that the reaction with bicyclic anhydrides did not work. Do the authors have a working hypothesis as to why this is? Obviously, the products of

these reactions would be very interesting, especially if stereochemistry could be controlled in the process.

Response: When we used bicyclic anhydrides and alkyl bromide substrates, it was found that this reaction provided decarbonylative product with very low yields. The mechanistic underpinnings for the change in reactivity between monocyclic and bicyclic anhydrides are not clear at this time.

Reviewer 1: In Figure 6c (bottom arrow), is the starting material 1b here (instead of 1a) or is there a mistake in the product structure?

Response: We have corrected this error by using **1b** instead of **1a** in the revised manuscript.

Conclusion: Overall, as stated above, this is a nice contribution to the field of Ni homoenolate chemistry and reductive cross-electrophile couplings and should be considered for publication in Nature Communications after minor revisions.

Reviewer 2 writes: The manuscript by Walsh, Mao et al. describes a nickel-catalyzed decarbonylative cross-electrophile coupling between cyclic anhydrides and alkyl bromides to afford alkyl carboxylic acids. The reaction is promoted using a catalytic system comprised of Ni(cod)₂, bipyridine and zinc dust in DMA at 80 °C, affording the desired acid products in moderate to good yields. The reaction is applicable to succinic and glutaric anhydrides as well as to primary and secondary alkyl bromides with tolerance to several functional groups. Mechanistic experiments indicated the intermediacy of a nickelalactone (“nickel homoenolate” in the case of a five-membered cycle) and an alkyl radical, while eliminating possible involvement of an alkylzinc bromide. Based on the experimental results and the known organometallic chemistry of nickel, a catalytic cycle featuring C-O oxidative addition of the anhydride to Ni(0) and subsequent decarbonylation as well as single electron transfer from Ni(I) to the alkyl bromide was proposed. The nickel(0)-mediated activation of cyclic anhydride and subsequent decarbonylation have been known for some time, as demonstrated by the previous works by Yamamoto and Rovis. That said, the present work is notable in that it has successfully translated this stoichiometric reactivity into a productive catalytic

process. The reaction would be useful as a method for the synthesis of carboxylic acids from readily available cyclic anhydrides and alkyl bromides. Publication of this manuscript in Nature Communications may be positively considered, while the authors are suggested to address the following points.

Response: We appreciate the reviewer's support and her/his recognition that this chemistry would be useful as a method for the synthesis of carboxylic acids from readily available cyclic anhydrides.

Some other comments:

Reviewer 2: The authors focus on the chemistry of homoenolates in the Introduction, devoting a lot of sentences and figures to discuss seminal works on homoenolates, NHC-derived homoenolates (Fig 1), metal homoenolates derived from cyclopropanol derivatives (Fig 2a,c), and organozinc homoenolates (Fig 2b). However, the reviewer feels that this introduction is not fully relevant, and would urge the authors to reconsider how the present work should be introduced. This is because the organonickel species generated from glutaric anhydride, while being referred to as "higher homologue" of homoenolate, has little to do with the chemistry discussed in the Introduction. Focusing solely on homoenolate in the Introduction looks irrelevant, especially given the substantial weight of glutaric anhydride (and its derivative) in the reaction scope. In fact, more than 70% of the products in Table 2 are derived from glutaric anhydride derivatives. The reviewer believes it would be possible to compose a more relevant introduction to this work from different contexts, for example, (Ni-catalyzed) C-O activation, decarbonylation, cross-electrophile coupling, carboxylic acid synthesis, and so on.

Response: We have removed NHC chemistry in the revised manuscript and focused on the catalytic generation of metal homoenolates and their higher homologues. What's more, palladium-catalyzed C-H activation to generate and functionalize palladium-homoenolate of carboxylic acid derivatives were added in the introduction. Based on the reviewer's suggestions, substituted succinic anhydrides (**1c-1f**) were synthesized to better clarify the application of homoenolates.

Reviewer 2: Page 4, "We have been interested in developing new transformations of homoenolates." The reviewer cannot see the relevance of one of the references cited for this statement (ref 42 by Wan et al). A clarification is necessary.

Response: We have removed this reference in the revised manuscript. The correct reference (Org. Lett. 2013, 15, 2298.) was added in the revised manuscript as ref 46.

Reviewer 2: The authors are suggested to explore substituted succinic anhydrides to better clarify the scope and limitation, and (if the reaction worked) the regioselectivity of C-O cleavage and C-C coupling.

Response: We have synthesized substituted succinic anhydrides (**1c-1f**), and explored the scope about those starting materials under standard conditions, which are summarized as below.

Reviewer 2: Does the reaction work using alkyl iodides or chlorides instead of alkyl bromides? Relevant comments would better clarify the scope of this reaction.

Response: When we used 1-chlorohexane to replace of 1-bromohexane as starting material, it was found only trace product under standard conditions. The 1-chlorohexane is not as active as 1-bromohexane, which resulted in the poor conversion. When we used 1-iodohexane instead of 1-bromohexane, affording **4bg** in 39% isolated yield. The sentence “In addition, when 1-chlorohexane or 1-iodohexane were used in place of 1-bromohexane, the AY were lower than 40% (see Supporting Information)” has been added in the revised manuscript.

Reviewer 2: Page 7, “...10.0 mmol of glutaric anhydride **1b** was coupled with 1-bromo-4-chlorobutane **2j**...” This should be corrected, as 15 mmol of glutaric acid and 10 mmol of alkyl bromide were used according to the Supporting Information. In addition, Fig 4 also needs correction because “**4bj** 10.0 mmol 1.55 g 87%” below the product structure is misleading.

Response: We have corrected this error in the revised manuscript. To test the scalability of this transformation, 15.0 mmol of glutaric anhydride **1b** was coupled with 1-bromo-4-chlorobutane **2j** under the standard conditions. The decarbonylative cross-coupled product **4bj** was isolated in 87% yield (1.55 g). (Fig. 3).

Fig. 3. Scale-up the transformation on 10.0 mmol scale

Reviewer 3 writes: This manuscript describes a Ni-catalyzed decarbonylative alkylation reaction of cyclic anhydrides. Upon oxidative addition to a cyclic anhydride, Ni undergoes decarbonylation to yield a key cyclic Ni-homoenolate intermediate that is subsequently alkylated to give beta- and gamma-alkylated carboxylic acids. Analogous reactions have been reported by Yamamoto and Rovis (ref. 43, 44) using stoichiometric Ni and zinc reagents. The authors used Zn as the terminal reductant to accomplish Ni turnover. The authors claim catalytic generation of homoenolate remains underdeveloped. In fact, the vast pool of reports on Csp³-H activation of carboxylic acids and their derivatives were ignored. The products involved in this manuscript can be readily prepared using beta- and gamma-Csp³-H alkylation protocols, on a much broader substrate scope. Therefore, novelty and synthetic potential of the current work is capped. The work might be publishable if the authors can go beyond six-membered cyclic anhydride to large ring size, which is surely outside of the scope of most C-H functionalization methods. Alternatively, if the authors can accomplish enantioselective alkylation using branched, racemic/meso anhydrides and chiral ligands, the manuscript would be significantly strengthened. Based on recent reports by Fu, it should be accomplishable.

Response: Recently, palladium catalyzed β -Csp³-H bond alkylation of carboxylic acid derivatives has witnessed significant process (*Chem. Soc. Rev.* **2019**, *48*, 4921). However, in the C-H bond functionalization protocol, a coordinating directing group needs to be preinstalled before the functionalization reaction and removed after the reaction. Furthermore, to the best of our knowledge, inexpensive nickel-catalyzed β -Csp³-H bond alkylation of carboxylic acid derivatives is extremely rare (*J. Am. Chem. Soc.* **2014**, *136*, 1789). Transition-metal-catalyzed γ -Csp³-H bond functionalization of carboxylic acids derivatives is a long-standing challenge, possibly due to the selectivity between β and γ positions. More recently, few examples based on the γ -Csp³-H bond functionalization of carboxylic acid derivatives have reported (*Angew. Chem., Int. Ed.* **2019**, *58*, 13773; *Angew. Chem., Int. Ed.* **2020**, DOI 10.1002/anie.202002362; *Angew. Chem., Int. Ed.* **2020**, DOI 10.1002/anie.202003271). To overcome the unwanted β -Csp³-H bond functionalized product, β all substituted carboxylic acids were required. Also, inexpensive nickel-catalyzed transformation is a limitation in this area.

Other issues

Reviewer 3: The introduction part needs major revision to include progress in the C-H activation paradigm and clearly show advantages of the current work.

Response: The strategy for directing group-assisted palladium-catalyzed β -C-H activation and palladium-catalyzed γ -functionalization of C-H's were added in the revised manuscript. However, at the present stage, C-H functionalization (especially for C-H alkylation) of aliphatic carboxylic acids without attaching exogenous auxiliary has been so far limited at the proximal β -position as well as γ -position. And the inexpensive nickel-catalyzed generation and functionalization of β -, or γ -nickel-bound carboxylic acid derivative remains underdeveloped. Our current work is complementary to the deficiency of previous reports.

Reviewer 3: Fig. 7, mechanistic cycle, is it possible that Zn reduces intermediate II to Ni(I) and the corresponding Ni(I) undergoes oxidative addition to alkyl-Br? This alternative pathway also gives III and radical mechanism can still operate.

Response: The reviewer's mechanistic proposal is possible, but we favor Ni(II) oxidative trapping of the alkyl radical for cross-electrophile coupling over Ni(I) oxidative addition to alkyl-Br based on the literature precedence. (*J. Am. Chem. Soc.* **2013**, *135*, 16192–16197. *J. Am. Chem. Soc.* **2014**, *136*, 17645–17651. *J. Am. Chem. Soc.* **2013**, *135*, 12004–12012. *J. Am. Chem. Soc.* **2014**, *136*, 1, 48–51. *Science*. **2014**, *345*, 433–436. *Science*. **2014**, *345*, 437–440. *J. Org. Chem.* **2014**, *79*, 4793–4798.)

Reviewer 3: Page 10, last paragraph, there are two “we” in the second sentence.

Response: We have corrected this error.

Reviewer 3: Substrates with ring size beyond glutaric anhydride need to be commented.

Response: We have synthesized large ring size anhydride **1j** and **1k** to screen the reaction. Unfortunately, they didn't give the desired cross-coupling products in moderate yields. As a result, we screened catalysts, ligands, solvents, reaction temperatures, reaction concentrations, reaction times and so on. However, the yields are very low. The corresponding results were shown below.

We are grateful to the referees for their careful review of our manuscript and their insightful comments that have helped to strengthen the manuscript.

REVIEWERS' COMMENTS

Reviewer #1 (Remarks to the Author):

In this revised manuscript, Walsh, Mao and coworkers have addressed all of my comments from the first round of peer-review (Reviewer 1). With these changes, the manuscript is suitable for publication in Nature Communications in my opinion.

Two minor comments/modifications prior to publication:

1. The quality of the plot in Figure 5c is quite poor (blurry for me). Please make sure a non-blurry plot is included in the final version of the manuscript.

2. I appreciate the control experiments the authors performed to address my question about the mechanistic possibility of " β -hydride elimination after CO elimination to generate the acrylate intermediate which could react in a radical conjugate addition with the alkyl bromide under reductive conditions". From what I can tell in this revised manuscript, a comment about these control experiments (on page 4 of the response to reviewer comments letter) is not included in the manuscript. I would highly recommend mentioning these experiments (and including them in the SI) in the final version of the manuscript since other readers may have the same question as I originally had about this mechanistic scenario.

Reviewer #2 (Remarks to the Author):

In the revised manuscript, the authors have carefully addressed this reviewer's comments on the original manuscript. The revised manuscript also seems to have resolved issues raised by other reviewers. The expanded substrate scope including substituted succinic anhydrides (Table 2) is notable, and further justified the presentation of this work as the chemistry of homoenolates and their homologues. The manuscript, with respect to its scientific contents, appears suitable for publication in Nature Communications.

Reviewer #3 (Remarks to the Author):

The authors have adequately addressed the technical issues raised by the reviewers. I like their responses regarding comparison with palladium homoenolates generated from C-H activation. Indeed, most reports used a directing auxiliary to accomplish the Pd-insertion. More importantly, generation of gamma-Pd enolates remains challenging. Additional mechanistic experiments are helpful. The revised manuscript has been improved and publication can be considered.

Minor issues:

1. Fig 1. should include an example of gamma-Pd enolates from C-H activation.

2. Reactivity of tertiary bromides should be commented.

3. Comments in Table 2 are ambiguous. Cyclic anhydrides used for "primary bromides" and "secondary bromides" should be indicated (succinic anhydride and glutaric anhydride). Otherwise readers can confuse with the section called "other anhydrides".

4. For the scale-up reaction, the amount of 2j (10 mmol) should be included in the text (15 mmol 1b and 10 mmol in the Fig 3 caption are confusing).

5. Table 3, what does "consumed" mean? For 4bg, it should be AY. For 2g, it should be the remaining

% since the alkyl bromide was consumed much slower than the cyclic anhydride. RT should be placed under the arrow, instead of in the footnotes. Why not run this reaction at higher temperature? Does this result mean the generation of alkyl radical is rate-limiting?

COMMENTS TO AUTHOR:

Reviewer 1: In this revised manuscript, Walsh, Mao and coworkers have addressed all of my comments from the first round of peer-review (Reviewer 1). With these changes, the manuscript is suitable for publication in Nature Communications in my opinion.

Response: We appreciate the reviewer's support and her/his recognition that the manuscript is suitable for publication in Nature Communications.

Reviewer 1 continues: The quality of the plot in Figure 5c is quite poor (blurry for me). Please make sure a non-blurry plot is included in the final version of the manuscript.

Response: We have made the 5-exo-trig experiment to evaluate ring-closed versus ring-opened ratios as a function of catalyst loading (Figure 5c) clearer in the revised manuscript.

Reviewer 1 continues: I appreciate the control experiments the authors performed to address my question about the mechanistic possibility of " β -hydride elimination after CO elimination to generate the acrylate intermediate which could react in a radical conjugate addition with the alkyl bromide under reductive conditions". From what I can tell in this revised manuscript, a comment about these control experiments (on page 4 of the response to reviewer comments letter) is not included in the manuscript. I would highly recommend mentioning these experiments (and including them in the SI) in the final version of the manuscript since other readers may have the same question as I originally had about this mechanistic scenario.

Response: We have added these control experiments in the revised manuscript.

Reviewer 2 writes: In the revised manuscript, the authors have carefully addressed this reviewer's comments on the original manuscript. The revised manuscript also seems to have resolved issues raised by other reviewers. The expanded substrate scope including substituted succinic anhydrides (Table 2) is notable, and further justified the presentation of this work as the chemistry of homoenolates and their homologues. The manuscript, with respect to its scientific contents, appears suitable for publication in Nature Communications.

Response: We appreciate the reviewer's support.

Reviewer 3 writes: The authors have adequately addressed the technical issues raised by the reviewers. I like their responses regarding comparison with palladium homoenolates generated from C-H activation. Indeed, most reports used a directing auxiliary to accomplish the Pd-insertion. More importantly, generation of gamma-Pd enolates remains challenging. Additional mechanistic experiments are helpful. The revised manuscript has been improved and publication can be considered.

Response: We appreciate the reviewer's support and her/his recognition that we have adequately addressed the technical issues raised by the reviewers.

Reviewer 3 continues: Fig 1. should include an example of gamma-Pd enolates from C-H activation.

Response: We added an example of gamma-Pd enolates from C-H activation in the revised manuscript.

Reviewer 3 continues: Reactivity of tertiary bromides should be commented.

Response: We have tried the tertiary bromides such as tert-butyl bromide and Ad-Br under standard conditions. Unfortunately, they did not generate the desired decarbonylative products.

Reviewer 3 continues: Comments in Table 2 are ambiguous. Cyclic anhydrides used for "primary bromides" and "secondary bromides" should be indicated (succinic anhydride and glutaric anhydride). Otherwise readers can confuse with the section called "other anhydrides".

Response: We have corrected this issue as the reviewer mentioned and updated the Table 2 in the revised manuscript.

Reviewer 3 continues: For the scale-up reaction, the amount of 2j (10 mmol) should be included in the text (15 mmol 1b and 10 mmol in the Fig 3 caption are confusing).

Response: We have corrected this issue in the revised manuscript.

Reviewer 3 continues: Table 3, what does "consumed" mean? For 4bg, it should be AY. For 2g, it should be the remaining % since the alkyl bromide was consumed much slower than the cyclic anhydride. RT should be placed under the arrow, instead of in the footnotes. Why not run this reaction at higher temperature? Does this result mean the generation of alkyl radical is rate-limiting?

Response: We used "AY" instead of "consumed" for 4bg, and used "remaining %" instead of "consumed" for 2g. We have placed "RT" under the arrow instead of "T (°C)" in the revised manuscript. We did run this reaction at higher temperature (80 °C), and the results are the same as RT (room temperature).

We thank the reviewers for their helpful suggestion. With the changes we have made, we hope that the manuscript is now acceptable for publication in Nature Communications.